# TSFF-Net: A deep fake video detection model based on two-stream feature domain fusion

**Hangchuan Zhang** [ID]**, Caiping Hu** [ID]*****, Shiyu Min** [ID]**, Hui Sui, Guola Zhou**

Department of Computer Engineering, Jinling Institute of Technology, Nanjing, Jiangsu, China

* hucp@jit.edu.cn

## Abstract

With the advancement of deep forgery techniques, particularly propelled by generative adversarial networks (GANs), identifying deepfake faces has become increasingly challenging. Although existing forgery detection methods can identify tampering details within manipulated images, their effectiveness significantly diminishes in complex scenes, especially in low-quality images subjected to compression. To address this issue, we proposed a novel deep face forgery video detection model named Two-Stream Feature Domain Fusion Network (TSFF-Net). This model comprises spatial and frequency domain feature extraction branches, a feature extraction layer, and a Transformer layer. In the feature extraction module, we utilize the Scharr operator to extract edge features from facial images, while also integrating frequency domain information from these images. This combination enhances the model's ability to detect low-quality deepfake videos. Experimental results demonstrate the superiority of our method, achieving detection accuracies of 97.7%, 91.0%, 98.9%, and 90.0% on the FaceForensics++ dataset for Deepfake, Face2Face, FaceSwap, and NeuralTextures forgeries, respectively. Additionally, our model exhibits promising results in cross-dataset experiments.. The code used in this study is available at: https://github.com/hwZHc/TSFF-Net.git.

**Data Availability Statement:** The code used in this study is available at: https://github.com/hwZHc/TSFF-Net.git This study utilizes two publicly available facial forgery datasets: FaceForensics++ and Celeb-DF. We strictly adhered to the terms and conditions set by the dataset providers to ensure

## 1. Introduction

With the rapid advancement of deep learning, the proliferation of fake images and videos generated by Deepfake [1] technology, which is based on deep generative methods, has become prevalent on the Internet. This widespread distribution poses a significant threat to personal privacy and social security. Consequently, researchers are actively exploring more effective methods to combat the evolving and accessible nature of deepfake techniques. Deepfake, a deep learning-based face forgery technique, alters facial images to morph one person's face into another, producing counterfeit videos in the process. Historically, video editing required specialized domain knowledge and tools, was time-consuming, and typically involved smaller-scale alterations, with the associated risks being underestimated. However, with the emergence of Deepfake technology and advancements in big data techniques, the accessibility of open and user-friendly forgery software and mobile applications has drastically reduced the complexity

that the data collection and analysis processes complied with all relevant guidelines. Additionally, all data have been properly anonymized, ensuring no personally identifiable information is included. The datasets used in this study are available at the following links: https://github.com/ondyari/FaceForensics https://github.com/yuezunli/celeb-deepfakeforensics.

**Funding:** This research was funded by Jinling Institute of Technology High-level Talent Research Start-up Project (jit-rcyj-202102)Key R&D Plan Project of Jiangsu Province (BE2022077). Jiangsu Province College Student Innovation Training Program Project (202313573080Y, 202313573081Y) and Jinling Institute of Technology Science and Education Integration Project (2022KJRH18) had no role in study design, data collection and analysis, decision to publish, or preparation of the manuscript.

**Competing interests:** The authors have declared that no competing interests exist.

of video editing. Moreover, the availability of large-scale public datasets has further facilitated the manipulation of facial images, exacerbating the issue.

Many existing methods for face forgery detection rely on Convolutional Neural Networks (CNNs) to learn forgery features from image frames within deepfake videos. However, due to the limited receptive field of the convolution process, these methods struggle to capture the global-local relationships across the entire image. Furthermore, as forged and authentic regions coexist within forgery images, exploiting the feature correlation between these regions can enhance the detection of sophisticated forgery techniques. Nevertheless, conventional CNN-based methods often fail to effectively capture this relationship. Additionally, in real-world scenarios, face forgery videos circulating on the internet are frequently highly compressed due to bandwidth limitations. Consequently, deep forgery detection models must possess cross-compression video detection capabilities to effectively identify forgeries in low-quality face videos.

To tackle these challenges, this paper combines CNN with Vision Transformer (ViT) [2] to construct a deep face forgery detection model featuring two-stream feature fusion across spatial and frequency domains. In contrast to existing Vision Transformer-based forgery detection methods, our approach utilizes EfficientNet [3] to extract feature information from both the spatial and frequency domains of face image frames within videos. This enables the model to exhibit the local modeling and anomaly detection capabilities of CNN when confronting deep forgery images, while also incorporating channel attention to enhance feature exchange between channels. Additionally, cross-attention is introduced to focus on the relationships between feature maps across different branches. Subsequently, the Transformer layer models the feature maps to generate classification tokens (CLS), and the classification results are obtained through a multi-layer perceptron head (MLP Head). The main contributions of this work are as follows:

1. We propose a deep forgery detection model, termed TSFF-Net, which combines CNN and ViT. This integration enables the model to effectively extract local features and perform global modeling, while reducing the time required for image feature extraction. As a result, the model's performance in detecting low-quality deepfake facial scenarios is significantly enhanced.

2. We design spatial and frequency stream branches to enhance the model's performance. In the spatial domain, we extract spatial features from the YCbCr color space, while in the frequency domain, we utilize Haar wavelets to extract frequency features. The cross-attention module within the Transformer learns the forgery details from both feature types, thereby improving the model's ability to comprehend the correlation between different feature domains and enhancing its generalization capability.

3. We integrate channel attention into the EfficientNet convolutional neural network to enhance the network's focus on significant feature channels, thereby improving the model's ability to detect potential forgery traces. Extensive experiments demonstrate that our model achieves state-of-the-art detection performance on both FaceForensics++ [4] and Celeb-DF [5] datasets.

## 2. Related works

### 2.1. Deep face forgery technology

Current techniques for face forgery detection can be broadly categorized into four types: FaceSwap [6], NeuralTextures [7], Face2Face [8], and Deepfake. FaceSwap involves replacing

one person's face with another, primarily using deep learning algorithms to detect and align faces in the image before replacing them, while preserving the original facial expressions and movements. NeuralTextures entails modifying various facial attributes of the target face, such as hair, skin color, age, and accessories. These methods typically integrate textual and spatial information alongside GAN-based techniques. Face2Face focuses on altering the expression of the target face, transforming expressions like "crying" into "laughing". In contrast, Deepfake generates synthetic faces that do not exist, leveraging techniques such as style transfer and textual information within the framework of GAN, distinguishing it from other face forgery methods mentioned above. Recent research [9] explores how to use GANs to synthesize high-quality deepfakes that can deceive existing forensic detectors. The researchers propose a method that can generate deepfake videos resistant to forensic analysis without the need for interaction with the detector. This method effectively evades detection by directly removing the indicative artifacts (i.e., GAN fingerprints) of GAN-generated images in the frequency spectrum. Article [10] focuses on adversarial attacks on fake-face detectors under both white-box and black-box scenarios. The researchers propose two GAN-based frameworks that can effectively reduce the detection accuracy of forensic models and have good transferability.

## 2.2. Face forgery detection methods based on deep learning

With the rapid development of deep forgery techniques based on adversarial generation, existing face forgery techniques have become increasingly adept at handling falsification traces such as tampering artifacts and boundary contours generated by image forgery, thereby rendering the generated face images and videos increasingly realistic. In this context, traditional face forgery detection techniques can no longer adequately address the task of detecting face forgery under unknown forgery techniques. Unlike traditional face forgery detection, CNN based on deep learning can extract pixel-level forgery features from face images, enabling effective detection of forged images and videos generated using GAN technology.

Currently, researchers have proposed some face forgery detection models based on traditional CNN, such as ResNet [11], XceptionNet [12], which have achieved good results in face forgery detection tasks. Haliassos et al. [13] proposed a lip forensic forged face detection method to address the variability of facial lip movements during face forgery. This method extracts a sequence of grayscale images of the face's lips from the face forgery video and passes them through a pre-trained ResNet for classification. Different from the detection of forged features on the face, Nirkin et al. [14] divided the face image into the face region and the external environment. They detected face forgery based on the face swapping technique by comparing the differences between these regions. In their approach, XceptionNet was used as the backbone and the face image was tightly segmented into the face part and the environment part, which were then fed into the model for training. Finally, the results were passed into the fully connected layer to obtain the output. The above image-based forgery detection methods have certain limitations when facing highly compressed internet videos. Researchers have found that the frequency domain information of images can still exhibit good characteristics after compression, and some forgery information that is not obvious in the RGB domain may be significant in the frequency domain. Qian et al. [15] designed a frequency domain perceptual decomposition module that performs a discrete cosine transform on the image to obtain the frequency domain features of the image, and proposed a local frequency domain information statistics module to obtain the frequency statistics features of the image's local regions. These two types of image information were then fed into a two-stream network with cross attention, F3-Net, for forgery detection. Li et al. [16] proposed an adaptive frequency feature generation module to extract frequency domain features from different frequency bands of

face video image frames. They combined the RBG and frequency domains of the images and fed them into the model for training respectively. To address the problem of inconsistent frequency feature distribution under different forgery methods, they used single center loss (SCL) to improve the intra-cluster compactness of real faces, thereby increasing the discrepancy between real and forged face videos.

In addition, compared to CNN networks, the Transformer is a network model that can efficiently process serialized information and has shown excellent performance in the field of natural language processing. In recent years, researchers have combined knowledge from the fields of computer vision (CV) and natural language processing (NLP) to develop ViT, which also performs well in image processing. Kong et al. [17] improved ViT by incorporating low-rank adaptation (LoRA) and single-center loss, optimizing the model to enhance detection accuracy while maintaining a low model complexity. This approach reduces the number of training parameters, conserving computational resources. Wang et al. [18] further advanced this line of work by combining the convolutional neural network MobileNet [19] with ViT, proposing the lightweight MobileViT model. This model effectively balances computational efficiency and detection accuracy, making it well-suited for mobile and resource-constrained environments. However, due to the large number of parameters in Transformer models, extracting features from images requires substantial computational power. Therefore, this paper combines the fast feature extraction capabilities of CNNs with the global information modeling strengths of Transformers, conducting experiments in low-quality, high-compression scenarios.

Unlike the aforementioned passive forgery detection techniques, the article [20] proposes an active defense measure to protect facial biometric information from the threat of deepfake counter-forensics attacks. Researchers have constructed a model with dual channels and multiple supervisors that can capture biometric features from various aspects. Through training, this method is able to purify counter-forensics images by eliminating adversarial perturbations.

While ViT achieves good performance in deep forgery detection tasks due to its strong global information modeling capability, traditional CNN-based detection methods focus more on the local information of the image. With the development of deep forgery techniques, the local artifact regions in images have become more subtle. Existing research often focuses on capturing these local features, but this approach tends to overlook the discrepancies between the real and forged regions in the forged video. Therefore, in this paper, we extracted the forgery features from the spatial and frequency domains of video image frames using CNN, respectively. We obtained the feature maps after weighting the channel information with the channel attention module and modeled the global information using the improved ViT and the cross-attention mechanism. This approach aimed to achieve deep forgery detection and enhance the model's performance in low-quality and high-compression-rate scenarios.

## 3. Method

In this paper, we propose a two-stream feature domain fusion model for deep forgery detection, integrating CNN with Transformer to leverage the efficient feature extraction capabilities of CNN. Our approach involves extracting local forged features from both spatial and frequency domains. The channel features are then weighted using the channel attention module and passed into the Transformer structure to capture the global correlation of these features. Additionally, we introduce the cross-attention mechanism to enhance the model's accuracy in scenarios involving cross-compression. The model structure, depicted in Fig 1, comprises a

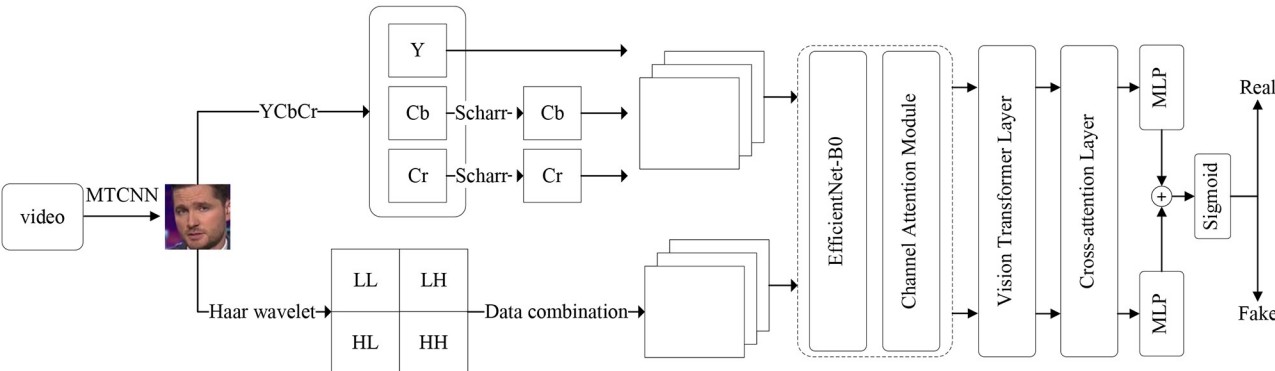

**Fig 1. Structure of deep forgery detection model with two-stream feature domain fusion.**

spatial domain feature extraction branch, a frequency domain feature extraction branch, an EfficientNet feature extraction layer, and a Transformer structure.

This study utilizes two publicly available facial forgery datasets: FaceForensics++ (FF++) and Celeb-DF. FaceForensics++ was released by a research team from the Technical University of Munich, primarily aimed at advancing video forgery detection techniques. Celeb-DF was provided by the University of Illinois at Chicago and is designed to offer more realistic deepfake video resources for forgery detection research. We strictly adhered to the terms and conditions set by the dataset providers to ensure that the data collection and analysis processes complied with all relevant guidelines. Additionally, all data have been properly anonymized, ensuring no personally identifiable information is included.

## 3.1. Spatial domain branching head

We utilize EfficientNet-B0 as the backbone network for the spatial-domain feature extraction branch, incorporating a spatial-domain color converter header before the backbone network. This header facilitates the conversion of extracted face images from the video's RGB color space to the YCbCr color space. This conversion enables the model to detect forged artifact features in the face images with greater sensitivity, thereby enhancing the accuracy of detecting various forging techniques.

The YCbCr color coding system is widely employed in various image processing tasks. It divides the image color information into two components: luminance (Y) and chrominance (Cb and Cr), as defined by Eq (1) (based on the ITU.BT-601 standard). This system enables the conversion of components from the RGB color space to the YCbCr color space.

$$\begin{bmatrix} Y \\ Cb \\ Cr \end{bmatrix} = \begin{bmatrix} 0.257 & 0.504 & 0.098 \\ 0.148 & 0.291 & 0.439 \\ 0.439 & 0.368 & 0.071 \end{bmatrix} \cdot \begin{bmatrix} R \\ G \\ B \end{bmatrix} + \begin{bmatrix} 16 \\ 128 \\ 128 \end{bmatrix} \tag{1}$$

During the process of fabricating a facial image, despite its natural appearance to the human eye, subtle forgery artifacts may persist in the chromaticity channel. These artifacts can be discerned by analyzing the chromaticity information of the image. Specifically, the chromaticity components, Cb (blue chromaticity) and Cr (red chromaticity), within the YCbCr color space exhibit heightened sensitivity to forged information resulting from image splicing. This heightened sensitivity proves advantageous in detecting forgeries such as face swapping and face splicing within the YCbCr color space.

To reduce the impact of redundant information in facial images, this paper employs an edge detection method to extract abnormal edge features. Inspired by Zhu et al. [21], the Scharr operator is integrated into the face forgery video detection process to enhance the discrepancy between filtered pixels. This improvement involves amplifying the weight coefficients within the filter, thereby facilitating better detection of forged facial edge features. Denoting the original image as $I$, and the gray values extracted by the edge detection algorithm detecting in the horizontal and vertical directions as $G_x$ and $G_y$ respectively, the operator is expressed in Eqs (2) and (3), where $I \in R^{H \times W \times C}$.

$$G_x(I) = \begin{bmatrix} -3 & 0 & 3 \\ -10 & 0 & 10 \\ -3 & 0 & 3 \end{bmatrix} \cdot I, \; G_y(I) = \begin{bmatrix} -3 & 10 & -3 \\ 0 & 0 & 0 \\ 3 & 10 & 3 \end{bmatrix} \cdot I \qquad (2)$$

$$G(I) = \sqrt{G_x^2 + G_y^2} \qquad (3)$$

We use the Scharr operator to extract edge feature from the image components of the Cb and Cr channels in the YCbCr color space. These features are then concatenated with the luminance component $Y$ along the image channel direction, as shown in Eq (4). Here, $Y_{I_o}$, $Cb_{I_o}$ and $Cr_{I_o}$ denote the YCbCr color components of the input image $I_o \in R^{224 \times 224 \times 3}$ after the transformation of Eq (1). The function $concat(\cdot)$ denote the concatenation of these matrices.

$$Y_{space} = concat\left(Y_{I_o}, G\left(Cb_{I_o}\right), G\left(Cr_{I_o}\right)\right) \qquad (4)$$

The facial images after YCbCr color space conversion and Scharr edge feature extraction are shown in Fig 2.

### 3.2. Frequency domain branching head

Detecting forgeries in low-quality face videos remains a significant challenge in current deep forgery detection research. Compression during the video encoding process leads to the loss of crucial facial information, including forgery details, making spatial-domain-based models ineffective for detecting numerous highly compressed videos available on the internet.

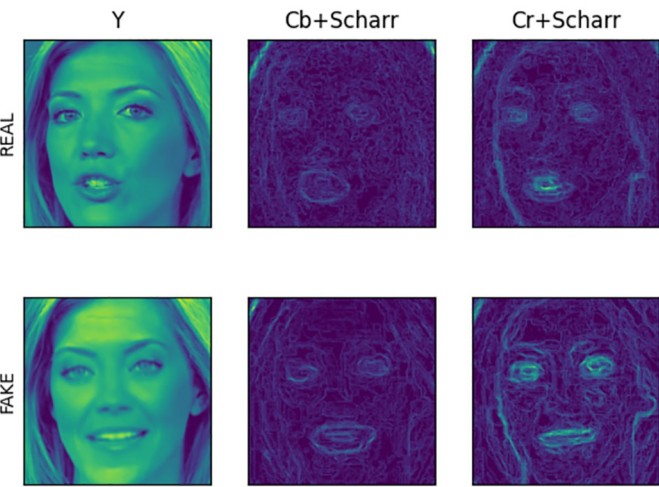

**Fig 2. YCbCr conversion and Scharr edge detection.**

However, significant features can still be discerned in the frequency domain of the images despite compression. To address this, we proposed a frequency domain transform head to convert spatial information into frequency information. This transformation enables the model to extract forged features from the frequency domain of the face image, integrate them with spatial domain features, and subsequently input them into the Transformer for forgery detection.

In the frequency domain branching head, similar to the work of WANG et al. [22], we choose the Haar wavelet transform to extract the frequency information in the image instead of the discrete cosine transform (DCT). After the image is transformed by Haar wavelet transform, its wavelet domain is divided into four subbands: image approximation coefficients, horizontal details, vertical details, and diagonal details. These subbands encompass not only the frequency domain components of the image, but also its spatial domain components, offering a more comprehensive representation compared to the DCT. Firstly, after obtaining the face image $I_o$, a grayscale transformation is performed to obtain $I_g$. Afterward, the Haar wavelet transform is applied to $I_g$ to obtain $LL$ (approximation coefficients), $LH$ (horizontal details), $HL$ (vertical details) and $HH$ (diagonal details) as shown in Eqs (5) and (6).

$$I_g = g(I_o) \tag{5}$$

$$\begin{bmatrix} LL & LH \\ HL & HH \end{bmatrix} = H^T I_g H \tag{6}$$

Where $g(\cdot)$ is the gray scale transformation function and $H = \frac{1}{\sqrt{2}} \begin{bmatrix} 1 & 1 \\ 1 & -1 \end{bmatrix}$ is the Haar transformation matrix.

After undergoing the Haar wavelet transformation, the elements in the $LL$ subband share the same value range as $I_g$, while the elements in the $LH$, $HL$, and $HH$ subbands have smaller value ranges. Our focus is primarily on the capturing horizontal, vertical, and diagonal details within the image's wavelet domain. To effectively utilize the frequency information obtained from the Haar wavelet transformation, we have designed a wavelet domain fusion module, as depicted in Eq (7).

$$Y_{fre} = \text{concat}\left(e^{\partial(LH)} + \partial(LL), e^{\partial(HL)} + \partial(LL), e^{\partial(HH)}\right) \tag{7}$$

Here, *concat*$(\cdot)$ represents the concatenation function of matrices, where $\partial(\cdot) = \frac{1-e^{-x}}{1+e^{-x}}$, aiming to confine the frequency information within the range of (0, 1). To better fuse the approximation coefficients, $LL$ is constrained to (0, 1) and then added to the detail coefficients $LH$ and $HL$. Regarding the frequency information, we observed that the elements of high-frequency information in Haar wavelets are too small. Therefore, an exponential transformation is applied to amplify the forged detail features, followed by concatenation along the image channel direction. Additionally, since the image size becomes one-fourth of the original after Haar wavelet transformation, bilinear interpolation is utilized to resize it to 224 × 224 before inputting it into the EfficientNet model. The visualization results of the original and forged face images after the frequency domain transformation are illustrated in Fig 3, where the forged traces are prominently visible.

## 3.3. EfficientNet

In this paper, we utilize EfficientNet to detect forged traces in videos. To enhance the model's training efficiency, we select EfficientNet-B0, the most lightweight model within the EfficientNet series, as the backbone network for feature extraction in both the spatial-domain and

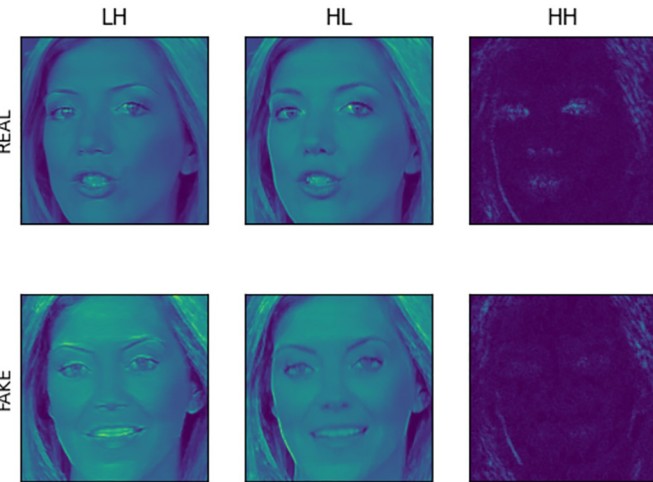

**Fig 3. Frequency domain transformation of face images.**

frequency-domain branches. Additionally, we improve the network's performance by incorporating a channel attention mechanism.

The original EfficientNet-B0 model incorporates a depth separable convolution MBConv (mobile inverted bottleneck conv) module to reduce computational parameters while integrating an SE (squeeze excitation) module between the depth and point convolutions. This SE module aids in learning the correlations between different feature channels, forming a channel-oriented attention convolution. However, the model fails to learn the implicit correlations between deep and shallow layers. Additionally, the SE module employs global average pooling (GAP) to grasp overall feature map information, thus being unable to capture anomalous features, such as falsified information. In contrast, global maximum pooling (GMP) captures maximum values, i.e., outliers, within the global range.

To address these limitations, our method incorporates an enhanced channel attention module that utilizes both global average pooling and global maximum pooling to facilitate soft attention weighting across the channel dimension. This module is integrated before the feature map output layer of EfficientNet-B0. The resulting attention weights of the global features are then fused with the feature map and subsequently fed into the Transformer for classification.

As shown in Fig 4, within the channel attention module, the input features, denoted as $X \in R^{H \times W \times C}$, undergo both global average pooling and global maximum pooling operations, resulting in two sets of features containing distinct spatial context information. Subsequently,

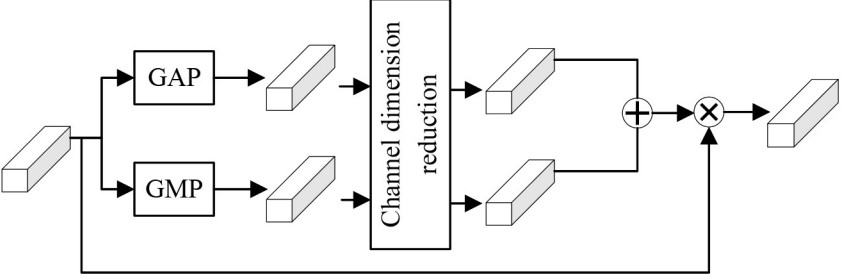

**Fig 4. Structure of channel attention.**

these two sets of global feature information are separately fed into fully connected and activation layers. This process facilitates channel dimension reduction followed by feature fusion, enabling the capture of richer global information features. Next, a linear layer is utilized to map the global feature information back to the initial channel dimension, thereby obtaining channel feature weights. Finally, the input features undergo weighted operations based on these channel feature weights.

### 3.4. Transformer

This paper adopts a combined approach using of CNN and ViT architectures to design a unified model, where features extracted from both spatial and frequency domains are passed into the Transformer structure for classification. In the original Vision Transformer (ViT) task, input images are divided into patches and flattened into a sequence of image blocks before being fed into the Transformer layers, leading to classification via fully connected layers. In contrast, our classification task involves the Transformer processing feature maps extracted from the spatial and frequency domains, rather than the original face images. Inspired by prior research [23], this paper enhances the Transformer encoder structure. For each feature block sequence $I \in R^{N \times C \times H \times W}$ outputted by the convolutional network, with each feature channel having a block size of $7 \times 7$ pixels, it undergoes a projection transformation before being passed into the encoder. Subsequently, the generated CLS token is concatenated with the feature block encoding sequence and fed into the branch cross-attention module.

The branch cross-feature attention mechanism is illustrated in Fig 5. Similar to the computation of self-attention [24], where $X_{cls}^s$ represents the CLS token acquired from the spatial domain branch feature blocks $X^s$, and $X^f$ denotes the feature sequence blocks of the frequency domain branch. After concatenating the two, corresponding key embeddings $k$ and value embeddings $v$ are obtained. During the computation, the spatial domain branch $X_{cls}^s$ needs to be resized to match the size of $X^f$ through projection, and the frequency domain branch follows the same procedure.

To better integrate feature information from different domains, cross-attention structures are employed to compute cross-attention between the feature information and CLS tokens output by the spatial and frequency domain encoders separately. In this paper, the Transformer encoder and cross-attention structures are alternately repeated four times. The CLS tokens output by the cross-attention module interact again with the feature block tokens in the next Transformer encoder, facilitating the interaction of feature information between different branches and enhancing the feature representation within each branch. After stacking four layers, both branches output CLS tokens. Following the multi-layer perceptron (MLP), the results from the two branches are summed to obtain the final classification result.

## 4. Experimental results

### 4.1. Datasets

To validate the effectiveness of our method, experiments were conducted on the FaceForensics ++ dataset and the Celeb-DF dataset. The FaceForensics++ dataset comprises 1,000 authentic facial videos from YouTube and 4,000 deepfake videos generated using various facial manipulation techniques, including Deepfake, Face2Face, FaceSwap, and NeuralTextures. Additionally, the FaceForensics++ dataset contains videos with three levels of compression quality: c0, c23, and c40, corresponding to raw videos, high-quality (HQ) videos, and low-quality (LQ) videos, respectively. Given the performance degradation of current face forgery detection methods when dealing with compressed videos, experiments in this paper were conducted

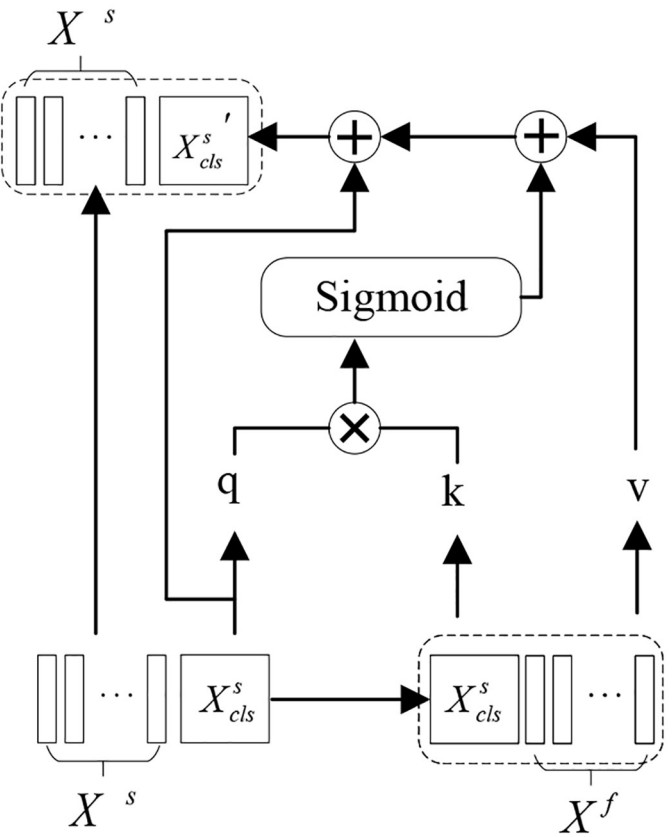

**Fig 5. Cross-attention structure.**

using both high-quality and low-quality versions of the FaceForensics++ dataset. The datasets of different qualities were divided into training, validation, and test sets in a ratio of 7:1:2.

The Celeb-DF dataset comprises 590 genuine videos and 5,639 forged videos. In comparison to the FaceForensics++ dataset, the Celeb-DF dataset employs more advanced facial manipulation techniques, which reduce visual artifacts in the videos, making the forged videos more challenging to detect. Since the Celeb-DF dataset does not disclose the specific forgery methods used, it is often utilized to test model performance in detecting facial manipulation videos with unknown forgery techniques. Therefore, we select the Celeb-DF dataset to evaluate our model's performance in cross-dataset scenarios. Table 1 below provides detailed information about the dataset used.

## 4.2. Experimental setting

For acquiring facial images from the facial manipulation videos, we employed Multi-task Cascaded Convolutional Networks (MTCNN) to extract facial regions from each frame of the

**Table 1. Basic information of the dataset used.**

| Dataset | Real(video) | Fake (video) | Real(Frame) | Fake(Frame) |
|---|---|---|---|---|
| FaceForensics++ | 1,000 | 4,000 | 509.9k | 2,039.6k |
| Celeb-DF | 590 | 5,639 | 225.4k | 2,116.8k |

**Table 2. Details of the training parameters.**

| Training Parameters | Settings |
|---|---|
| Optimizer | SGD |
| Image Resolution | 256x256 pixels |
| Learning Rate | 0.01 |
| Weight Decay | 0.000001 |
| Batch Size | 32 |

input videos. In both the training and validation sets, we uniformly sampled every 30 frames from each facial video. Similarly, for the test set, we selected every 30 frames from each video, resizing the facial images to 224 × 224 pixels. Additionally, we applied common data augmentation techniques, including horizontal flipping, Gaussian blurring, and Gaussian noise, to the original images during experimentation. The backbone networks in both the spatial and frequency domains utilize the EfficientNet-B0 model pre-trained on the ImageNet [25] dataset. We employed binary cross-entropy as the primary loss function, which is particularly well-suited for binary classification tasks. This loss function effectively reduces the difference between the true labels and the predicted values throughout the training process. The model was trained for 50 epochs, with performance on the validation set evaluated at the end of each epoch. To avoid the model getting stuck in local minima during later stages of training, we implemented an early stopping strategy. This strategy automatically halts training if the validation loss fails to improve for 10 consecutive epochs. The model was optimized using Stochastic Gradient Descent (SGD) with a learning rate of 0.01, weight decay of 0.000001, and a batch size of 32. Table 2 below summarizes the training parameters:

We evaluate the model's performance using two metrics: classification accuracy (ACC) and the area under the receiver operating characteristic curve (AUC). Both ACC and AUC are commonly used evaluation metrics in image classification tasks and are widely employed in existing research on deep forgery detection.

## 4.3. Experimental results

This paper presents comparative experiments on two datasets with different compression qualities, FF++ (HQ) and FF++ (LQ), to evaluate the detection performance of the proposed method. In Table 3, TSFF-Net achieves high detection accuracies of 97.67% for Deepfake and 98.96% for FaceSwap on the lightly compressed version of FF++ (HQ), outperforming other methods. However, its performance on Face2Face and NeuralTextures datasets is relatively lower at 91.0% and 90.0%, respectively, possibly due to the smaller tampered areas in these methods, limiting the detection of forged traces. Despite this, TSFF-Net demonstrates superior performance overall, particularly excelling in Deepfake and FaceSwap detection compared to

**Table 3. Accuracies of FaceForensics++ (HQ) (ACC).**

| Model | Deepfake (%) | Face2Face (%) | FaceSwap (%) | NeuralTextures (%) |
|---|---|---|---|---|
| SPSL [26] | 93.5 | 86.0 | 92.3 | 76.8 |
| CAG-MobileViT [18] | 93.7 | 94.1 | 96.3 | 87.9 |
| XceptionNet [4] | 96.4 | 86.9 | 90.3 | 80.7 |
| Fre-Xception [22] | 95.5 | 93.5 | 95.2 | 90.4 |
| IR-Capsule [27] | 97.3 | 81.8 | 94.2 | 79.3 |
| TSFF-Net (ours) | 97.7 | 91.0 | 98.9 | 90.0 |

**Table 4. Accuracies of FaceForensics++ (LQ) (ACC).**

| Model | Deepfake (%) | Face2Face (%) | FaceSwap (%) | NeuralTextures (%) |
|---|---|---|---|---|
| LTW [28] | 69.1 | 65.7 | 62.5 | 58.5 |
| RECCE [29] | 72.9 | 66.4 | 61.3 | 57.1 |
| Low-Rank ViT [17] | 73.5 | 63.8 | 65.3 | 54.2 |
| Muti-clues [30] | 75.3 | 64.4 | 64.1 | 59.5 |
| TSFF-Net (ours) | 85.1 | 83.1 | 78.5 | 66.0 |

other methods. Further improvements are needed to enhance detection accuracy for Face2-Face and NeuralTextures, particularly in capturing subtle forgery details.

Table 4 presents the ACC experimental results of various methods under the high compression scenario of FF++ (LQ). Our proposed TSFF-Net achieves notable accuracies of 85.1% for Deepfake and 83.1% for Face2Face, leveraging both spatial and frequency domains to extract richer features for forgery detection. However, the model exhibits lower accuracy on FaceSwap (78.5%) and NeuralTextures (66.0%) datasets, attributed to the loss of facial forgery information during compression and the subtle tampering details in these methods. Despite these challenges, TSFF-Net demonstrates robust performance compared to other methods, particularly excelling in detecting Deepfake and Face2Face forgeries under high compression scenarios, highlighting its effectiveness in cross-compression rate forgery detection.

Considering the efficacy of cross-dataset detection in our model, we integrate features from both spatial and frequency domains for feature selection. Utilizing the cross-attention mechanism, we capture regional inconsistencies among features and compressed artifacts, thereby enhancing the model's detection performance across different datasets. Table 5 presents the cross-dataset detection AUC results of TSFF-Net and other methods on the Celeb-DF dataset. Specifically, our model, trained on the FF++ (HQ) dataset, achieves a superior AUC score of 77.1%. Compared to CNN-Transformer and Fre-Xception, which also leverage frequency domain features for forgery detection, our model shows improvements of 9.29% and 4.83% in AUC score, respectively. This indicates the robust generalization capability and competitiveness of our approach.

## 4.4. Ablation experiments

The TSFF-Net comprises primarily spatial-domain and frequency-domain feature extraction branches, an enhanced EfficientNet feature extraction layer, and a Transformer layer with cross-attention. To validate the efficacy of the branch combination in our model, we conducted a comparative analysis of various branch configurations on the FF++ (HQ) dataset to assess their detection performance across different forgery scenarios.

**Table 5. Results of cross-database experiments on the Celeb-DF dataset (AUC).**

| Model | AUC (%) |
|---|---|
| CNN-Transformer [32] | 67.81 |
| Fre-Xception [22] | 72.27 |
| Face X-ray [31] | 74.76 |
| CAG-MobileViT [18] | 75.50 |
| SPSL [26] | 76.88 |
| TSFF-Net (ours) | 77.10 |

**Table 6. Ablation study results for branching strategies (ACC).**

| Model | Deepfake (%) | Face2Face (%) | FaceSwap (%) | NeuralTextures (%) |
|---|---|---|---|---|
| YCbCr | 81.0 | 76.0 | 56.7 | 61.9 |
| Freq | 85.0 | 90.7 | 95.9 | 58.8 |
| YCbCr + Freq | 96.0 | 89.7 | 98.9 | 83.9 |
| YCbCr + Freq + Cross-Attention | 97.7 | 91.0 | 98.9 | 90.0 |

The experimental results are presented in Table 6, where different configurations are evaluated:

- "YCbCr": Model using only a spatial-domain feature extraction branch, the unimproved EfficientNet structure, and a Transformer layer.

- "Freq": Model utilizing only a frequency-domain feature extraction branch, the unimproved EfficientNet structure, and a Transformer layer.

- "YCbCr + Freq": Model combining both spatial-domain and frequency-domain feature extraction branches, the unimproved EfficientNet structure, and a Transformer layer with cross-attention.

- "YCbCr + Freq + Cross-Attention": Model incorporating both spatial-domain and frequency-domain feature extraction branches, an enhanced EfficientNet feature extraction layer, and a Transformer layer with cross-attention.

From Table 6, it is evident that the model utilizing frequency-domain features (model "Freq") significantly outperforms the spatial-domain model ("YCbCr"), particularly on the Face2Face and FaceSwap datasets. This suggests that frequency-domain features are highly effective for detecting Face2Face and FaceSwap forgeries. However, the "Freq" model shows lower performance on the NeuralTextures dataset, indicating limitations in capturing the intricacies of this forgery method. The integrated model ("YCbCr + Freq") that combines both spatial and frequency-domain feature extraction branches demonstrates marked improvements across all forgery types, particularly boosting accuracy for Deepfake and NeuralTextures forgeries. This highlights the benefit of leveraging both spatial and frequency information for a more comprehensive detection approach.

Our final model ("YCbCr + Freq + Cross-Attention"), which includes an enhanced EfficientNet feature extraction layer and a cross-attention mechanism in the Transformer, achieves the highest detection performance across all datasets. This model records accuracy rates of 97.7% for Deepfake, 91.0% for Face2Face, 98.9% for FaceSwap, and 90.0% for NeuralTextures. The cross-attention mechanism significantly enhances the model's capability to integrate and utilize feature information from both domains, resulting in superior detection performance. These results demonstrate the efficacy of our TSFF-Net model in accurately detecting various types of face forgeries by effectively combining spatial and frequency domain information with advanced feature extraction and attention mechanisms.

## 5. Conclusions

In this paper, we propose TSFF-Net, a deep forgery detection model integrating an enhanced EfficientNet and Vision Transformer to improve detection accuracy across varying compression rates. By applying a YCbCr transformation, the model extracts forged details from the

chrominance channel and utilizes the Scharr operator for abnormal edge detection in face images. Incorporating Haar wavelet transformation enhances frequency domain feature extraction, which contributes to effective forgery detection. TSFF-Net leverages EfficientNet-B0 for efficient local feature extraction and introduces an advanced channel attention module to enhance feature extraction across deep and shallow levels. Through a cross-attention mechanism, TSFF-Net integrates spatial and frequency domain features to achieve robust forgery detection.

Experimental results on FF++ and Celeb-DF datasets demonstrate TSFF-Net's superior performance in cross-compression and cross-dataset scenarios, particularly excelling against Deepfake forgery methods under varying compression rates. However, challenges arise in detecting NeuralTextures due to its nuanced forgery effects across different facial regions, posing difficulties in capturing inter-region connections crucial for global information modeling. Further refinement is essential to adapt our method to the diverse array of internet-based video and image manipulation techniques, beyond those represented in current public datasets.

Future work will focus on developing more efficient and accurate detection methods, especially for low-quality forged images, to tackle the complexities of face video forgery detection in real-world scenarios.

## Author Contributions

**Conceptualization:** Caiping Hu.

**Data curation:** Guola Zhou.

**Investigation:** Hangchuan Zhang.

**Project administration:** Shiyu Min.

**Validation:** Hui Sui.

**Writing – original draft:** Hangchuan Zhang.

**Writing – review & editing:** Caiping Hu.

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
