## [Decision Letter · Decision Letter 0]

1 Sep 2024

PONE-D-24-22578TSFF-Net: A Deep Fake Video Detection Model Based on Two-Stream Feature Domain FusionPLOS ONE

Dear Dr. Hu,

Thank you for submitting your manuscript to PLOS ONE. After careful consideration, we feel that it has merit but does not fully meet PLOS ONE’s publication criteria as it currently stands. Therefore, we invite you to submit a revised version of the manuscript that addresses the points raised during the review process.

We look forward to receiving your revised manuscript.

Kind regards,

Feng Ding

Academic Editor

PLOS ONE

Journal Requirements:

2. In your Methods section, please include additional information about your dataset and ensure that you have included a statement specifying whether the collection and analysis method complied with the terms and conditions for the source of the data.

 This research was funded by Jinling Institute of Technology High-level Talent Research Start-up Project (jit-rcyj-202102)、Key R&D Plan Project of Jiangsu Province (BE2022077)、Jiangsu Province College Student Innovation Training Program Project (202313573080Y, 202313573081Y) and Jinling Institute of Technology Science and Education Integration Project (2022KJRH18).  

Additional Editor Comments :

Please carefully revise the manuscript to solve all the issues raised by reviewers. Also note that, it is not necessary to cite the papers mentioned by the reviewers unless you think it could be academically beneficial for your paper.

Reviewers' comments:

Reviewer's Responses to Questions

**Comments to the Author**

1. Is the manuscript technically sound, and do the data support the conclusions?

Reviewer #1: Yes

Reviewer #2: Yes

2. Has the statistical analysis been performed appropriately and rigorously? 

Reviewer #1: Yes

Reviewer #2: Yes

3. Have the authors made all data underlying the findings in their manuscript fully available?

Reviewer #1: Yes

Reviewer #2: Yes

4. Is the manuscript presented in an intelligible fashion and written in standard English?

Reviewer #1: Yes

Reviewer #2: Yes

5. Review Comments to the Author

Reviewer #1: The paper presents a novel deep forgery video detection model called Two-Stream Feature Domain Fusion Network (TSFF-Net), which aims to improve deep forgery detection in complex scenes and low-quality images. The experimental results are impressive, showing the high detection accuracy of the method on the FaceForensics++ dataset. But there are still some details worth discussing:

1. Many of the citations had problems showing up as 'Error! Reference source not found', such as the citations in the INTRODUCTION section for the datasets FaceForensics++ and Celeb-DF.

2.The proposed method uses efficient-B0 as backbone, has it tried to use other cnn structures such as ResNet or DenseNet?

3.The goal of the proposed method is to detect the latest emergence of Gan-based face forgery, but the datasets FaceForensics++ and Celeb-DF both seem to be based on encoder-decoder structures, not traditional GAN. hopefully the article has a discussion on gan-generated deepfake, for example, [1][2][3]

[1] Synthesizing black-box anti-forensics deepfakes with high visual quality

[2] Adversarial attack on fake-faces detectors under white and black box scenarios

[3] Securing facial bioinformation by eliminating adversarial perturbations

Reviewer #2: In this paper, the authors introduce a novel deep face forgery video detection model named Two-Stream Feature Domain Fusion Network (TSFF-Net), which enhances the model's ability to detect low-quality deepfake videos. Experimental results demonstrate the superiority of this method. In general, the idea of this paper technically makes sense; and the manuscript is easy to follow. However, the following issues should be addressed:

1. The introduction section should further compare the recent research work, especially the research based on transformer and hybrid models, such as RTSMFFDE-HKRR: A fault diagnosis method for train bearing in noise environment.

2. Identify and list research gaps in the related work section. Discuss how the proposed approach fills these gaps and offers advantages.

3. The model structure diagram in Figure 1 can further refine the specific structure of each branch.

4. Include a more detailed discussion of model training information, such as loss functions, hyperparameter optimization procedures, etc.

5. Provide details of the image dataset and training parameters in tabular form.

6. PLOS authors have the option to publish the peer review history of their article (what does this mean?). If published, this will include your full peer review and any attached files.

Reviewer #1: No

Reviewer #2: No

---

## [Author Response · Author response to Decision Letter 0]

11 Sep 2024

Original Manuscript ID: PONE-D-24-22578 

Original Article Title: “TSFF-Net: A Deep Fake Video Detection Model Based on Two-Stream Feature Domain Fusion”

To: Editor

Re: Response to reviewers

Dear Editor,

Thanks for your letter and review work on our manuscript entitled “TSFF-Net: A Deep Fake Video Detection Model Based on Two-Stream Feature Domain Fusion”. We greatly appreciate the reviewers’ complimentary comments and suggestions. These comments are very helpful and valuable for improving and polishing our paper. The point-by-point responses to reviewers’ comments are given in the following part. 

We have added additional information about the datasets in the Methods section and included a statement confirming that our data collection and analysis methods comply with the terms and conditions set by the data sources.

We confirm that our submission contains all the raw data required to replicate the results of the study, in accordance with PLOS's definition of the minimal data set. Specifically, the code used in this study is publicly available at: https://github.com/hwZHc/TSFF-Net.git, and all relevant data, metadata, and methods are provided within the manuscript and its Supporting Information files. This ensures that all study findings can be fully replicated.

Best regards,

Hangchuan Zhang, Caiping Hu, Shiyu Min, Hui Sui and Guola Zhou

Reviews and Responses

Reviewer #1:

Point 1: Many of the citations had problems showing up as 'Error! Reference source not found', such as the citations in the INTRODUCTION section for the datasets FaceForensics++ and Celeb-DF.

Response 1: In response to the "Error! Reference source not found" issue you mentioned, we have conducted a thorough examination and correction. We found that the problem occurred in the citation formatting of the reference literature, especially regarding the FaceForensics++ and Celeb-DF datasets mentioned in the "Introduction" section. We failed to format these citations correctly, which resulted in the citations not being displayed properly in the final document. Therefore, we rechecked the format of all citations to ensure they comply with the journal's citation standards. Particularly for the FaceForensics++ and Celeb-DF datasets, we have ensured the correctness of the citation format.

Point 2: The proposed method uses efficient-B0 as backbone, has it tried to use other cnn structures such as ResNet or DenseNet?

Response 2: Thank you for your insightful comment. In this study, we chose EfficientNet-B0 as the backbone network primarily because it offers a good balance between performance and efficiency. However, we did consider using other CNN architectures, such as ResNet and DenseNet. In preliminary experiments, we tested ResNet and DenseNet but found that their computational complexity was higher, leading to significantly increased training times. Although they could potentially provide slightly higher accuracy in certain cases, we believe that EfficientNet-B0 offers a better trade-off between accuracy and efficiency.

That being said, this does not mean that ResNet or DenseNet cannot be used for this task. In future research, we may further explore these or other more complex network architectures to enhance the model's performance.

Point 3: The goal of the proposed method is to detect the latest emergence of Gan-based face forgery, but the datasets FaceForensics++ and Celeb-DF both seem to be based on encoder-decoder structures, not traditional GAN. hopefully the article has a discussion on gan-generated deepfake, for example, [1][2][3].

[1] Synthesizing black-box anti-forensics deepfakes with high visual quality

[2] Adversarial attack on fake-faces detectors under white and black box scenarios

[3] Securing facial bioinformation by eliminating adversarial perturbations

Response 3: We appreciate your concern that while our datasets, FaceForensics++ and Celeb-DF, are based on encoder-decoder architectures, they do not represent the traditional GAN structures. We have chosen these datasets because they provide a diverse and high-quality set of deepfake samples, which is essential for assessing the performance of detection algorithms. Regarding the detection of GAN-generated deepfakes, we have included a discussion on this topic in Related works of our manuscript.

Reviewer #2:

Point 1: The introduction section should further compare the recent research work, especially the research based on transformer and hybrid models, such as RTSMFFDE-HKRR: A fault diagnosis method for train bearing in noise environment.

Response 1: We appreciate your suggestion to compare recent research work, we have discussed these models in detail in the Related Work section, we will revise the Introduction section to briefly introduce these models and provide a concise comparison with our research.

Point 2: Identify and list research gaps in the related work section. Discuss how the proposed approach fills these gaps and offers advantages.

Response 2: We have added relevant descriptions in section 2.2. At the end of Section 2.2, we elaborate on the shortcomings of existing research, which tends to focus more on capturing local features. However, this often overlooks the differences between genuine and forged areas in forged videos. Therefore, we combine facial image features in both spatial and frequency domains, using an improved Vision Transformer (ViT) and cross-attention mechanism for global information modeling, to achieve deepfake detection and enhance the model's performance in scenarios with low quality and high compression rates.

Point 3: The model structure diagram in Figure 1 can further refine the specific structure of each branch.

Response 3: Thank you for your suggestion. We acknowledge that the model structure diagram (Figure 1) could provide clearer information by further refining the specific structure of each branch. In the revised version, we will update Figure 1 to illustrate the detailed structure of each branch. This will help to better visualize the proposed model architecture and the role of each branch.

Point 4: Include a more detailed discussion of model training information, such as loss functions, hyperparameter optimization procedures, etc.

Response 4: We have added relevant descriptions in section 4.2.

Point 5: Provide details of the image dataset and training parameters in tabular form.

Response 5: Thank you for your question. Below is a table that provides details of the image dataset used and the training parameters:

Table 1. Basic information of the dataset used.

Dataset Real(video) Fake (video) Real(Frame) Fake(Frame)

FaceForensics++ 1,000 4,000 509.9k 2,039.6k

Celeb-DF 590 5,639 225.4k 2,116.8k

Table 2. Details of the training parameters.

Training Parameters Settings

Optimizer SGD

Image Resolution 256x256 pixels

Learning Rate 0.01

Weight Decay 0.000001

Batch Size 32

---

## [Decision Letter · Decision Letter 1]

17 Sep 2024

TSFF-Net: A Deep Fake Video Detection Model Based on Two-Stream Feature Domain Fusion

PONE-D-24-22578R1

Dear Dr. Hu,

We’re pleased to inform you that your manuscript has been judged scientifically suitable for publication and will be formally accepted for publication once it meets all outstanding technical requirements.

Kind regards,

Feng Ding

Academic Editor

PLOS ONE

Additional Editor Comments (optional):

Reviewers' comments:

Reviewer's Responses to Questions

**Comments to the Author**

1. If the authors have adequately addressed your comments raised in a previous round of review and you feel that this manuscript is now acceptable for publication, you may indicate that here to bypass the “Comments to the Author” section, enter your conflict of interest statement in the “Confidential to Editor” section, and submit your "Accept" recommendation.

Reviewer #1: All comments have been addressed

Reviewer #2: (No Response)

2. Is the manuscript technically sound, and do the data support the conclusions?

Reviewer #1: Yes

Reviewer #2: (No Response)

3. Has the statistical analysis been performed appropriately and rigorously? 

Reviewer #1: Yes

Reviewer #2: (No Response)

4. Have the authors made all data underlying the findings in their manuscript fully available?

Reviewer #1: Yes

Reviewer #2: (No Response)

5. Is the manuscript presented in an intelligible fashion and written in standard English?

Reviewer #1: (No Response)

Reviewer #2: (No Response)

6. Review Comments to the Author

Reviewer #1: All my questions were answered.

Reviewer #2: For Recommendation 1, it is necessary to add new literature and train-related literature in the references.

7. PLOS authors have the option to publish the peer review history of their article (what does this mean?). If published, this will include your full peer review and any attached files.

Reviewer #1: No

Reviewer #2: No

---

## [Editor Report · Acceptance letter]

20 Sep 2024

PONE-D-24-22578R1 

PLOS ONE

Dear Dr. Hu, 

I'm pleased to inform you that your manuscript has been deemed suitable for publication in PLOS ONE. Congratulations! Your manuscript is now being handed over to our production team.

Kind regards, 

on behalf of

Dr. Feng Ding 

Academic Editor

PLOS ONE